# Legend or Truth: Mature CD4^+^CD8^+^ Double-Positive T Cells in the Periphery in Health and Disease

**DOI:** 10.3390/biomedicines11102702

**Published:** 2023-10-05

**Authors:** Magdalena Hagen, Luca Pangrazzi, Lourdes Rocamora-Reverte, Birgit Weinberger

**Affiliations:** Institute for Biomedical Aging Research, University of Innsbruck, 6020 Innsbruck, Austria

**Keywords:** adaptive immunity, CD4^+^CD8^+^ T cells, aging, T cell development, T cell differentiation, viral infections

## Abstract

The expression of CD4 and CD8 co-receptors defines two distinct T cell populations with specialized functions. While CD4^+^ T cells support and modulate immune responses through different T-helper (Th) and regulatory subtypes, CD8^+^ T cells eliminate cells that might threaten the organism, for example, virus-infected or tumor cells. However, a paradoxical population of CD4^+^CD8^+^ double-positive (DP) T cells challenging this paradigm has been found in the peripheral blood. This subset has been observed in healthy as well as pathological conditions, suggesting unique and well-defined functions. Furthermore, DP T cells express activation markers and exhibit memory-like features, displaying an effector memory (EM) and central memory (CM) phenotype. A subset expressing high CD4 (CD4^bright+^) and intermediate CD8 (CD8^dim+^) levels and a population of CD8^bright+^CD4^dim+^ T cells have been identified within DP T cells, suggesting that this small subpopulation may be heterogeneous. This review summarizes the current literature on DP T cells in humans in health and diseases. In addition, we point out that strategies to better characterize this minor T cell subset’s role in regulating immune responses are necessary.

## 1. Introduction

Mature T cells have been initially described by the mutually exclusive expression of either CD4 or CD8 co-receptors. These molecules allow the definition of two major T cell populations, each destined to fulfill distinct functions [1]. CD4^+^ and CD8^+^ expressing T cells are key players in establishing and maintaining immune responses [2]. Both subsets express a unique surface receptor called T cell receptor (TCR), created by the somatic rearrangement of DNA and chain pairing. The expression of TCRs allows these cells to recognize a broad spectrum of pathogens, tumors, and other threats [2,3]. Antigenic stimulation of the TCR leads to CD4^+^ and CD8^+^ T cell proliferation, differentiation, and the release of effector cytokines [4]. However, despite their similarities, once an antigen has been recognized, these two major T cell subsets fulfill entirely different functions. While the major role of CD4^+^ T cells, so-called T helper cells, is to support and regulate the responses of other immune cells, CD8^+^ T cells, so-called cytotoxic T cells, act on their own to directly clear a potential danger by eliminating, for example, virus-infected or tumor cells [5,6]. Although the expression of either CD4 or CD8 co-receptors determines the function of a mature T cell and rules out the other, the existence of a T cell population in the periphery expressing CD4 and CD8 simultaneously (i.e., CD4^+^CD8^+^ double-positive, DP, T cells) has been described [7]. This unconventional mature T cell population was found in the peripheral blood of various species in varying proportions, including human (~3%), swine (30–55%), chicken (20–40%), cynomolgus monkey (~5%), and rat (~6%) [8,9,10,11,12]. In addition, in mice, a distinct DP T cell population representing a significant part of the intraepithelial lymphocytes (IEL), the first line of defense against dietary antigens and microbes underneath the intestinal epithelial cells, has been described [13]. While the presence of this remarkable DP T cell subset in human peripheral blood has been documented primarily in several pathological conditions such as viral infections and cancer, this population can also be detected in healthy individuals in whom they increase with age [7,14,15]. Thus, it appears that this “paradigm-breaking” T cell population may fulfill roles that are distinct from their conventional single-positive (SP) counterparts. In addition, this minuscule T cell population was even shown to consist of two distinct subsets, that is, a population expressing high CD4 (CD4^bright+^) and intermediate CD8 (CD8^dim+^) levels and a population of CD8^bright+^CD4^dim+^ T cells [16,17]. Despite this, the differences and similarities between these two subpopulations have been poorly explored so far.

## 2. Discovery of Mature CD4^+^CD8^+^ Double-Positive T Cells in the Periphery

Despite the broadly accepted “mutually exclusive CD4-CD8 paradigm”, a small proportion of T cells outside the thymus express the co-receptors CD4 and CD8 simultaneously. This minor mature CD4^+^CD8^+^ DP T cell population in human blood was first documented in the 1980s using two-color fluorescence flow cytometry [8]. According to Blue et al., approximately 3% of T cells derived from the blood of healthy donors express both co-receptors [8]. One decade later, Ortolani et al. discovered that CD4^+^CD8^+^ DP T cells comprise two subsets, CD4^bright+^CD8^dim+^ and CD4^dim+^CD8^bright+^ T cells [16], which was later confirmed by Nascimbeni et al. [17]. In the blood of 10 healthy donors, they observed that CD4^dim+^CD8^bright+^ T cells make up for 0.15% to 0.9% of peripheral blood mononuclear cells (PBMCs), while 0.2% to 8.2% of total PBMCs are CD4^bright+^CD8^dim+^ T cells [17]. In addition to the peripheral blood, DP T cells were found in substantial numbers in the human small intestinal lamina propria and the human adult liver [18,19]. In lymphocytes isolated from liver specimens of 15 donors, 5.5% DP T cells could be found within total T cells, in contrast to 1.3% in the blood of the same donors [19]. Intriguingly, within the human small intestinal lamina propria, Abuzakouk et al. detected, on average, even 14% CD4^+^CD8^+^ DP T cells [18]. Therefore, DP T cells appear to form a significant component of the human lamina propria [18].

While there is already a considerable donor-to-donor variation in the proportion of CD4^+^CD8^+^ DP T cells, age, sex, and body mass index (BMI) were additionally shown to influence the percentages of DP T cells in the blood. A study examining blood samples of a Spanish cohort showed that healthy adults had higher percentages of CD4^+^CD8^+^ DP T cells than children [20]. On average, the proportion of DP T cells was two times higher in adults than in children [20]. In agreement with this finding, both DP T cell populations, CD4^bright+^CD8^dim+^ as well as CD4^dim+^CD8^bright+^ DP T cells, were shown to be more prevalent in the blood of healthy older adults compared to young and middle-aged individuals, potentially reflecting long exposures to chronic antigenic stimulation such as cytomegalovirus (CMV) [15]. In addition to the impact of age, higher percentages of CD4^+^CD8^+^ DP T cells were found in women compared to men in a Colombian cohort [14]. However, this finding could not be confirmed in other cohorts [20,21]. Furthermore, the proportion of DP T cells might depend on external factors like nutrition, as it was shown that the percentage of DP T cells positively correlates with BMI [20].

## 3. Potential Origin of Mature CD4^+^CD8^+^ Double-Positive T Cells

T cells originate in the bone marrow, from which the immature T cell precursors migrate to the thymus, where they develop into mature T cells [2]. According to a well-accepted paradigm, successful T cell development conventionally results in two populations of mature T cells, CD4^+^ T helper cells, and cytotoxic CD8^+^ T cells [22]. Cytotoxic T cells recognize antigens presented by major histocompatibility complex (MHC) class I molecules with their TCR and with their CD8 co-receptors also binding to MHC I proteins. In contrast, T helper cells recognize MHC class II-peptide complexes with their CD4 co-receptors binding to MHC II molecules [23].

T cell progenitors leaving the bone marrow and entering the thymus still lack the expression of CD4 and CD8 molecules (Figure 1) [2]. These CD4^-^CD8^-^ DN thymocytes proliferate in the thymus, differentiate into immature CD4^+^CD8^+^ DP thymocytes, and undergo T cell receptor rearrangement [2,24]. Cortical epithelial cells in the thymus express the antigen-presenting molecules, MHC class I and MHC class II, which are loaded with self-peptides [3]. DP thymocytes poorly interacting with these self-peptides bound to MHC molecules undergo death by neglect. At the same time, too strong interactions induce apoptotic cell death of these autoreactive T cell progenitors [3]. Thymocytes exhibiting a desired intermediate TCR-binding strength classically proceed with lineage-specific differentiation into exclusively CD4^+^ or CD8^+^ expressing SP mature T cells [3]. While immature CD4^+^CD8^+^ DP T cells stimulated by MHC class II molecules differentiate into CD4^+^ SP T helper cells, MHC class I stimulated DP T cells differentiate into CD8^+^ SP cytotoxic T cells [25]. The development into CD4^+^ SP T cells is driven by the transcription factor ThPOK, while expression of Runx3 drives CD8^+^ SP T cell development [26]. According to current knowledge, mature naïve T cells leaving the thymus are supposed to express either CD4 or CD8 exclusively and are destined to fulfill distinct functions. Nevertheless, the existence of CD4^+^CD8^+^ DP T cells in the periphery has been documented decades ago and, to date, several studies corroborate this finding.

The aforementioned raises the question of whether these cells may escape from the thymus in an immature form or, alternatively, represent a subset of genuinely mature T cells. Intriguingly, aging is accompanied by the altered development and functionality of various cell types of the immune system [27]. In particular, aging results in a substantial regression of the size of the thymus and severe deterioration of the thymic structure, leading to a drastic reduction of mature T cell output [28,29]. The fact that DP T cells increase with age despite this substantial involution of the thymus contrasts with the theory that DP thymocytes might escape from the thymus in an immature state. In agreement with this, CD4^+^CD8^+^ DP T cells were shown to contain low levels of T cell receptor excision circles (TREC) and have short telomeres, indicating a differentiated phenotype [17]. In addition, they were shown to express molecules typically produced by effector cells, including granzyme B, perforin, and IFN-γ [7]. Moreover, DP T cells lack the expression of CD1a, further indicating that these cells are fully mature [17].

Therefore, CD4^+^CD8^+^ DP T cells in the periphery may derive from their SP counterparts rather than having egressed from the thymus as immature T cells [7]. Indeed, Sullivan and colleagues stimulated PBMCs in vitro with phytohemagglutinin (PHA) and found that this process induced an intermediate (dim) expression of CD4 on CD8 T cells [30]. The authors suggested that CD4^dim+^CD8^bright+^ T cells showed the typical phenotype of activated CD8^+^ T cells, expressing high levels of CD95, CD25, CD38, and CD69 [30]. In vivo studies performed by Imlach et al. confirmed these findings and suggested that activated CD8^+^ T cells expressing CD4 are a target for human immunodeficiency virus (HIV) [31]. On the other hand, it has been shown that terminally differentiated peripheral CD4^+^ T cells may acquire the expression of CD8, therefore generating CD4^bright+^CD8^dim+^ DP T cells [32,33]. In mice, it was shown that CD4^+^ SP T cells can acquire the expression of CD8 upon entry into the intestinal epithelial layer [34]. This cell type exhibits a reduced expression of ThPOK and gained the expression of Runx3, which induces CD8 expression driven by TGF-β and retinoic acid present in the gut [26,35]. Intriguingly, these DP T cells express CD8αα homodimers instead of CD8αβ [34,36]. In agreement with this, CD4^bright+^CD8^dim+^ DP T cells in the human blood, which may derive from CD4^+^ SP T cells, express CD8αα, while CD4^dim+^CD8^bright+^ DP T cells potentially deriving from CD8^+^ SP T cells express CD8αβ receptors [7,37].

**Figure 1 biomedicines-11-02702-f001:**
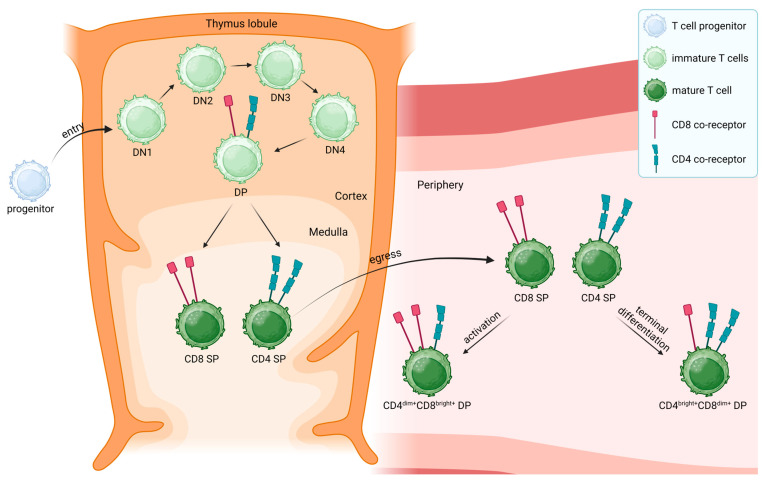
T cell development and potential origin of mature CD4^dim+^CD8^bright+^ and CD4^bright+^CD8^dim+^ double-positive (DP) T cells. T cell progenitors (light blue) originating from the bone marrow enter the thymus (orange) as double-negative (DN) T cells, which lack the expression of both CD4 and CD8 co-receptors, to undergo positive and negative selections [2,3,38]. Immature DN T cells (light green) go through four stages of differentiation (DN1 to DN4) in which rearrangement of the T cell receptor (TCR) β-chain and the expression of a preTCR expression occur [3]. DN4 T cells undergo rearrangement of the α-chain of the TCR, leading to the transition to immature DP T cells, which express a rearranged TCR and both CD4 (red) and CD8 (turquoise) co-receptors [3]. DP T cells that recognize antigens presented via MHC class II receptors differentiate into mature CD4^+^ single-positive (SP) T cells (dark green), while recognition of antigens presented by MHC class I molecules results in mature CD8^+^ SP T cells, which leave the thymus into the periphery [25]. Despite the paradigm that peripheral T cells exclusively express either CD4 or CD8, mature CD4^+^CD8^+^ DP T cells were found outside the thymus [8]. These DP T cells can further be divided into two subsets, CD4^dim+^CD8^bright+^ and CD4^bright+^CD8^dim+^ T cells, respectively [16]. On the one hand, it was suggested that the activation of CD8^+^ SP T cells may lead to the surface expression of CD4, resulting in CD4^dim+^CD8^bright+^ T cells [30]. On the other hand, CD8 expression was suggested to be acquired by CD4^+^ SP T cells upon terminal differentiation, leading to a CD4^bright+^CD8^dim+^ T cell population [32,33]. Created with BioRender.com (accessed on 29.09.2023).

## 4. CD4^+^ and CD8^+^ Single-Positive T Cells

CD4^+^ and CD8^+^ SP T cells fulfill different functions in adaptive immune responses. On the one hand, CD4^+^ SP T cells play a crucial role in coordinating adaptive immunity by supporting and regulating other immune cells. In particular, they are essential for the production of specific antibodies by B cells, they can support and inhibit the cytotoxic response of CD8^+^ T cells, and they regulate the function of macrophages and other innate immune cells [5]. Naïve CD4^+^ T cells in the periphery can differentiate into effector cells, which reduce the surface expression of the co-stimulatory receptors CD28 and CD27 and the chemokine receptor CCR7 after repeated cycles of activation [39,40]. Differentiated CD4^+^ T cells can be divided into different subtypes showing distinct functions depending on their cytokine production. The main CD4^+^ T helper subtypes are Th1, Th2, Th17, T-follicular helper (Tfh) cells, and regulatory T cells (Tregs) [5]. Th1 cells are characterized by the production of the cytokine IFN-γ and support cell-mediated immunity and phagocyte-dependent immune responses [5,41]. The signature cytokines of Th2 cells are IL-4, IL-5, and IL-13, and they mediate phagocyte-independent immune responses [5,41]. Th17 cells produce the cytokine IL-17, playing a key role in the defense against microbial infections [5,42]. Tfh cells are crucial for antibody production by B cells when interacting with them in the germinal centers [43]. Finally, Tregs are essential to maintain peripheral tolerance and to prevent autoimmunity and chronic inflammation [44]. They produce TGF-β and IL-10 and are characterized by the expression of the surface marker CD25 and the transcription factor FoxP3 [44].

In contrast, the ultimate purpose of CD8^+^ SP T cells is to eliminate cells infected with intracellular pathogens and tumors [6]. After encountering antigens, naïve CD45RA^+^CD28^+^CCR7^+^CD8^+^ T cells can differentiate into CD45RA^-^CD28^+^CCR7^+^ central memory (CM) T cells and CD45RA^-^CD28-CCR7^-^ effector memory (EM) T cells [45,46,47]. CD28 is a co-stimulatory molecule which, upon binding to its ligands CD80 and CD86 expressed by antigen-presenting cells, induces the production of IL-2, a cytokine required for T cell proliferation and differentiation into effector and memory T cells [48,49]. CD45RA, an isoform of CD45, is a key player in the initiation of TCR signaling [50]. Compared to EM CD8^+^ T cells, CM CD8^+^ T cells are more cytolytic towards their target cells and express the chemokine receptor CCR7 for homing to secondary lymphoid organs, while EM CD8^+^ are negative for the marker CCR7 [46]. Terminally differentiated effector memory (TEMRA) T cells are characterized by the lack of CD28 and CCR7 and the expression of CD45RA [45]. In addition, after repeated antigen exposure, CD8^+^ T cells acquire the expression of the terminal differentiation marker CD57 [51]. Exhaustion of CD8^+^ T cells is regulated by the expression of co-inhibitory molecules, such as programmed cell death-1 (PD-1) and cytotoxic T-lymphocyte-associated protein 4 (CTLA-4) [52]. Intriguingly, chronic infection with CMV leads to a substantial expansion of EM T cells and TEMRA CD8^+^ T cells [53].

## 5. Peripheral CD4^+^CD8^+^ Double-Positive T Cells

Although several studies have investigated the phenotype of CD4^+^ and CD8^+^ SP T cells, little is known about the phenotype and function of CD4^+^CD8^+^ DP T cells in the periphery. As mentioned above, memory (EM and CM) T cells are characterized by the lack of CD45RA expression [46,47]. A study conducted by Waschbisch et al. revealed that approximately 54% of DP T cells in the blood exhibit an effector memory T cell phenotype, while about 8% show a central memory phenotype [54]. In more detail, according to Nascimbeni et al., CD4^dim+^CD8^bright+^ T cells mainly exhibit a CCR7^-^CD45RA^-^ EM T cell phenotype, while CD4^bright+^CD8^dim+^ T cells are predominantly CCR7^+^CD45RA^-^ CM T cells [17]. CD4^+^CD8^+^ DP T cells are generally more differentiated than their SP counterparts [17]. In addition, the terminal differentiation marker CD57 is found more frequently on DP T cells than on CD4^+^ SP T cells, suggesting that DP T cells may show at least some features of terminally differentiated T cells [17]. In a recent study, the frequency of the activation marker CD154 was shown to increase in CD4^dim+^CD8^bright+^ T cells, in comparison with the other T cell populations [14]. Despite this, a detailed phenotypical characterization of both CD4^+^CD8^+^ DP T cell subsets has yet to be performed.

An essential factor for effector T cell function is their capability to migrate between the blood, peripheral tissues, and secondary lymphoid organs [55]. T cell migration is mediated by the expression of diverse chemokine receptors and selectins [55]. CM T cells typically move between secondary lymphoid organs and the blood, while EM T cells circulate in the blood and home to inflamed tissues [55]. In concordance with the memory-like phenotype of peripheral DP T cells, they express less of the lymph node homing marker CD62L compared to their SP counterparts [17]. In addition, CD4^+^CD8^+^ DP T cells isolated from human blood were shown to express the tissue-homing marker CXCR3 [17]. So far, no studies have assessed the migratory capacity of CD4^dim+^CD8^bright+^ and CD4^bright+^CD8^dim+^ T cells. To better understand the role of these subsets in immune responses, it is necessary to decipher the migratory abilities of DP cells.

Regarding the role of DP T cells in the gut, studies in murine IELs suggest that CD4^+^CD8αα^+^ DP T cells may fulfill regulatory functions, evident by their production of the anti-inflammatory cytokines IL-10 and TGF-β [34,56].

## 6. Peripheral CD4^+^CD8^+^ Double-Positive T Cells in Disease

As already mentioned, DP T cells are increased in the blood of healthy older adults compared to young ones [15], and it has been observed that older adults with higher DP T cell numbers had some associated comorbidities [57]. DP T cell populations are altered in different ailments, namely (1) infectious diseases, (2) inflammatory/autoimmune diseases, and (3) cancer.

CD4^+^CD8^+^ DP T cells were found to be increased in patients with viral infections such as HIV and COVID-19, indicating a potential role of this minor T cell population in the clearance of viruses [32,58,59,60,61]. In patients infected with hepatitis C, increased numbers of CD4^bright+^CD8^dim+^ DP T cells were documented [62]. High percentages of DP T cells were also found in patients suffering from chronic Chagas disease caused by the parasite Trypanosoma cruzii [63]. In some cases, the DP T cell population is suggested as a marker to assess the severity of a disease, as is the case for hemorrhagic fever with renal syndrome (HFRS) [64]. In patients suffering from Dengue viral infection, the frequency of DP T cells is significantly increased in individuals at risk of developing plasma leakage and can be therefore used as a marker for the disease progression [65].DP T cells are also increased in different autoimmune diseases such as multiple sclerosis (MS) and rheumatoid arthritis (RA) and in patients suffering from Sjögren’s syndrome [66,67,68,69,70]. The presence of DP T cells is also proposed as a marker for severity in several inflammatory diseases. In systemic lupus erythematosus (SLE), peripheral DP cells are associated with the risk of developing renal impairment [71]. In addition, the expansion of DP T cells in RA is associated with joint damage and frequent escalation of therapy [70]. Another example of DP T cell presence in inflammation is the development of graft versus host disease (GVHD), which has been recently associated with the appearance of a DP T cell population that was originally not present in the graft [72].In the context of cancer, CD4^+^CD8^+^ DP T cells were found to infiltrate cutaneous T cell lymphomas and were increased in nodular lymphocyte predominant Hodgkin lymphoma, breast cancer, and hepatocellular carcinoma [73,74,75,76,77]. Other studies have identified DP T cells in human melanoma, which originate from TCR-stimulated CD4^+^ or CD8^+^ SP cells [78] and may potentiate antitumor responses via helper functions of CD4^dim+^CD8^bright+^ DP cells [79].

Controversial findings regarding the function of this aforementioned T cell subset have been published in the last decades. In the 1990s, two individual groups suggested that CD4^dim+^CD8^bright+^ T cells exhibit an activated phenotype, while CD4^bright+^CD8^dim+^ T cells appeared more differentiated based on the expression of CD57 [16,80]. Both groups proposed, however, that CD4^dim+^CD8^bright+^ T cells may be associated with EBV-infectious mononucleosis and may disappear after a short time [16,80]. In patients with symptomatic HIV infection, CD4^+^CD8^+^ DP T cells exhibited increased levels of activation and exhaustion based on their expression of the activation marker CD38 and the exhaustion marker PD-1 [81]. Along the same lines, the expression of the exhaustion markers PD-1 and Tim-3 has been differentially identified on CD4^bright+^CD8^dim+^ and CD4^dim+^CD8^bright+^ T cells, respectively [82]. In addition, DP T cells in HIV-infected subjects produced high levels of the pro-inflammatory cytokines IFN-γ and TNF [32,60,81]. Similarly, CD4^+^CD8^+^ DP T cells from healthy donors were shown to produce both cytokines when stimulated with virus-infected cell lysates [17]. In vitro studies demonstrated that CD4^+^CD8^+^ DP T cells of acutely infected patients and “HIV controllers” (patients who control viral replication without antiretroviral therapy [83]) have an increased proliferative capacity in response to HIV proteins compared to their SP counterparts [84]. Similar to DP T cells in viral infections, DP T cells exhibited an activated phenotype and produced IFN-γ in chronic Chagas disease [63]. 

In patients who suffer from RA, DP T cells display a memory T cell phenotype and produce increased levels of the cytokines IL-12 and IL-4 in comparison to DP T cells of healthy donors [69]. In hepatocellular carcinoma patients, DP T cells exhibit an activated and exhausted phenotype and produce IFN-γ and TNF upon stimulation, resembling what is described for HIV-infected individuals [77]. Furthermore, CD4^bright+^CD8^dim+^ cells infiltrating human cutaneous T cell lymphoma mediate specific MHC class I-restricted cytotoxic activity toward fresh autologous tumor cells and exhibit enhanced cytokine expression, proliferation, and cytotoxic activity in response to CMV and HIV-1 antigens [85]. In addition, DP T cells from urological tumors have been suggested to display immunoregulatory functions triggering Th2 polarization while restricting Th1 in CD4^+^ SP cells [86]. Moreover, a recent study has identified a population of DP Tfh cells that play a role in the regulation of humoral immunity in chronic inflammatory lesions [87], suggesting the existence of novel unknown functions from DP T cells.

## 7. Conclusions and Future Directions

In summary, information already present in the literature suggests that peripheral CD4^+^CD8^+^ DP T cells may play an important role in regulating immune responses in pathological and physiological conditions. It has been reported that this subpopulation represents a T cell subset with a distinct and mature phenotype. Whether DP T cells may additionally be present within lymphoid organs (i.e., the spleen or lymph nodes) is currently unknown and should be assessed. Furthermore, the existence of the minor mature CD4^bright+^CD8^dim+^ and CD4^dim+^CD8^bright+^ DP T cell populations in the periphery is nonnegligible and different studies suggest that these cells potentially fulfill distinct functions. Although several reports investigated CD4^+^CD8^+^ DP T cells in pathological conditions, our knowledge regarding human mature CD4^+^CD8^+^ DP T cells in the blood of healthy donors is still incomplete. As findings in human diseases point towards a distinct role of this minor T cell population compared with other T cell subsets, the study of the presence of this DP population under physiological conditions is of high interest. Therefore, an in-depth characterization of their phenotype and their function is required to shed some light on their enigmatic role in healthy humans. In particular, studies intensively investigating cytokine production, proliferative capacity, and migratory behavior as well as potential suppressive (regulatory) capacities of mature CD4^bright+^CD8^dim+^ and CD4^dim+^CD8^bright+^ DP T cells in the periphery in physiological conditions are recommended and may deliver essential knowledge on these highly interesting T cell populations. 

## Data Availability

No new data were created or analyzed in this study. Data sharing is not applicable to this article.

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
