# Peer review of "Legend or Truth: Mature CD4+CD8+ Double-Positive T Cells in the Periphery in Health and Disease"

_biomedicines, 2023, doi:10.3390/biomedicines11102702_

Round 1

Reviewer 1 Report

GENERAL COMMENTS:

Hagen et al have reviewed the current knowledge about mature CD4+CD8+ double positive (DP) T cells in humans -a minor population in blood- whose characteristics and functional roles are little known at present. This is an interesting topic, because all the (preliminary) evidence point out that these cells could play a crucial role in certain physiological and pathological conditions, and the review is well supported by an appropriate bibliography. However, the manuscript present deficiencies that authors should address adequately, as described below.

SPECIFIC COMMENTS FOR REVISION:

MAJOR COMMENTS:

1.- I suggest the authors to include a new section to summarize in a specific part what we currently know about the functional role of each of the CD4+CD8+ DP T cells in physiological conditions (e.g., after Section 5), also including the potential role of CD4bright/CD8dim T cells in antitumor immunosurveillance (please see: doi:10.1002/adbi.202200169).

2.- The role of DP T cells in disease should be significantly extended and updated, and improved in its structure, to try to describe the functions that each DP T-cell subset is proposed to have by disease categories (e.g.: ordered by infectious diseases, autoimmune and inflammatory diseases, tumors and others.)

MINOR COMMENTS:

3.- Ref 1 is not appropriate to support the sentence “Mature T cells have been initially described by the mutually exclusive expression of 24 either CD4 or CD8 co-receptors. These molecules allow the definition of two major T cell 25 populations, each destined to fulfill distinct functions”. Please replace by the original paper or by a high-impact review.

4.- Figure 1 should be cited in the main text (now there is no reference to figure).

5.- Pg 6 line 2: “bright” should be written as superscript.

Author Response

MAJOR COMMENTS:

1.- I suggest the authors to include a new section to summarize in a specific part what we currently know about the functional role of each of the CD4+CD8+ DP T cells in physiological conditions (e.g., after Section 5), also including the potential role of CD4bright/CD8dim T cells in antitumor immunosurveillance (please see: doi:10.1002/adbi.202200169).

We thank Reviewer#1 for her/his constructive suggestions. We included information on the role of DP T cells in antitumor immunosurveillance using the suggested reference. (lines 289-293)

2.- The role of DP T cells in disease should be significantly extended and updated, and improved in its structure, to try to describe the functions that each DP T-cell subset is proposed to have by disease categories (e.g.: ordered by infectious diseases, autoimmune and inflammatory diseases, tumors and others.)

We have modified section 6 (CD4+CD8+ double-positive T cells in disease) in which we summarize the presence and potential roles of DP T cells (and the CD4bright+CD8dim+ and CD4dim+CD8bright+ subpopulations). We have rewritten this section including newer bibliography as well as structuring the text according to the kind of disease described. (lines 233-297)

MINOR COMMENTS:

3.- Ref 1 is not appropriate to support the sentence “Mature T cells have been initially described by the mutually exclusive expression of 24 either CD4 or CD8 co-receptors. These molecules allow the definition of two major T cell 25 populations, each destined to fulfill distinct functions”. Please replace by the original paper or by a high-impact review.

We have replaced the reference by a higher-impact one. (line 26)

4.- Figure 1 should be cited in the main text (now there is no reference to figure).

We are sorry for this mistake. Now the reference to Figure 1 is included in the main text. (line 97)

5.- Pg 6 line 2: “bright” should be written as superscript.

This mistake is now corrected in the revised manuscript. (line 228)

Reviewer 2 Report

The work of Hagen and colleagues entitled ‘ Legend or truth: mature CD4+CD8+ double-positive T cells in the periphery in health and disease ‘ is written as a short review. The main topic of this review, the existence, origin and function of peripheral DP T lymphocytes, is very timely and to some extent overlooked. Authors describe these unusual peripheral cells, point towards the existence of subtypes CD4brightCD8dim and CD4dimCD8bright, and correctly note the inconsistence of the results presented in the literature. In general, the manuscript exhibits some inconsistency, repetitions (e.g., the fact that peripheral DP lymphocytes increase with age is mentioned three times) and poor structure of the text. Authors write 23 lines on thymic development which is only loosely linked to the rest. Much shorter introduction to thymic DP cells would be sufficient and the focus on thymus escape which may be relevant to the main topic is missing. Almost a page (from less than 6 pages) focuses on the description of single positive peripheral T cells, but, again, only a minor part is required for the text on peripheral DP T cells. Shortening and focus on key DP-related issues are needed. But, most importantly, discussed of recent literature is completely missing! The whole work is written in line with an article from 2004 (ref. 10). However, there are several key findings in current works on peripheral DP T cells (e.g., refs.[1]) which would help to answer (at least partially) several unresolved questions mentioned in the text. It is critically important to discuss these works in the review and, instead of naming surface receptors or other markers without any link to the potential function of these cells, try to guide a reader to some conclusions, which are in line with a current knowledge about these super-interesting immune cells. In conclusion, the review should focus on peripheral DP T cells, their subsets, origin, potential function …. not insufficiently describe other immunological topics (e.g. thymic development).

Few things to include:

·         Origin of peripheral DP T cells – thymic escape or differentiation from SP T cells, or both.

·         Origin of the two DP T cell subtypes (not subclasses).

·         CD8alphaalpha or CD8alphabeta reappears in DP T cells – their function is different.

·         Why aging leads to the increase in peripheral DP T cells. Reinfections?

·         Can proportion of these cells be really so high as observed in the ref. 10 – up to 8% of total blood T cells in one donor. Was this donor sick?

·         Future perspective. Key questions for now.

Minor comments:

Line 29: TCR doesn’t undergo RANDOM chain pairing. It’s pretty non-random process.

Line 40: missing references

Lines: 59-60: Where are these numbers (%) coming from? I could not find it in the cited work – there is  % range in Fig. 1.

[1] Schad et al., “Tumor-Induced Double Positive T Cells Display Distinct Lineage Commitment Mechanisms and Functions”; Hess et al., “Inflammatory CD4/CD8 Double-Positive Human T Cells Arise from Reactive CD8 T Cells and Are Sufficient to Mediate GVHD Pathology”; Clénet et al., “Peripheral Human CD4+CD8+ T Lymphocytes Exhibit a Memory Phenotype and Enhanced Responses to IL-2, IL-7 and IL-15”; Bohner et al., “Double Positive CD4+CD8+ T Cells Are Enriched in Urological Cancers and Favor T Helper-2 Polarization.”

Some sentences need revision for clarity.

Author Response

Thanks to the constructive suggestions of Reviewer#2 we were able to improve our manuscript by significantly expanding the description of mature DP T cells by approximately 700 words. We agree about the length of the thymic development, but we consider the whole concept of key importance to understand why DP T cells in the periphery are unexpected. In addition, disagreement among the reviewers’ comments made us decide to keep that part as it originally was. As suggested by Reviewer#2 we did remove repetition on the impact of aging on DP T cells. Further we added 13 recent (from 2018-2023) references to the manuscript (lines 347-348, 403-404, 407-408, 447-449, 464-467, 479-484, 499-500, 509-512, 518-524) and highlighted that CD4dim+CD8bright+ have an activated phenotype as reported by Gonzalez-Mancera et al. (lines 215-217).

Few things to include:

  • Origin of peripheral DP T cells – thymic escape or differentiation from SP T cells, or both.

As suggested the chapter describing the origin of mature CD4+CD8+ double-positive T cells in the periphery was expanded. (lines 137-144)

  • Origin of the two DP T cell subtypes (not subclasses).

This was corrected. (lines 9, 173 and 174)

  • CD8alphaalpha or CD8alphabeta reappears in DP T cells – their function is different.

Information of CD8αα and CD8αβ receptors was added to the manuscript. (lines 140-144 and 230-232)

  • Why aging leads to the increase in peripheral DP T cells. Reinfections?

Unfortunately, we do not have a clear explanation. However, it is speculated that the increase of DP T cells with age may reflect long exposure to chronic antigenic stimulation like for example in the case of CMV infection à this information was added to the manuscript. (lines 79-82)

  • Can proportion of these cells be really so high as observed in the ref. 10 – up to 8% of total blood T cells in one donor. Was this donor sick?

According to the article of Nascimbeni et al. the donor with 8.2% CD4bright+CD8dim+ DP T cells of total PBMCs was healthy. (lines 63-65)

  • Future perspective. Key questions for now.

Particular suggestions for future perspectives have been added to the manuscript. (lines 313-317)

Minor comments:

Line 29: TCR doesn’t undergo RANDOM chain pairing. It’s pretty non-random process.

The word random was removed from the manuscript.

Line 40: missing references

We are sorry for this mistake. The missing references were added.

Lines: 59-60: Where are these numbers (%) coming from? I could not find it in the cited work – there is  % range in Fig. 1.

We report % ranges in the revised manuscript (in the previous version of the manuscript an average was calculated).

Reviewer 3 Report

This review article is well written and conveys authors’ scientific messages on the functional characterization of the mature CD4+CD8+ double positive (DP) T cells in the periphery in health and disease. I have a couple of additional comments on current form of manuscript. First, as described also by the authors, these DP T cells in the peripheral blood are a mature and heterogeneous population and may have distinct functions in both normal and pathological conditions. The descriptions of summarized current knowledge on the mature DP T cells present in the periphery sound nicely stated. Apart from the periphery, another mature DP T cells are located in the intestinal compartment, particularly, as a part of the intraepithelial lymphocytes (IELs) underneath the intestinal epithelial cells. These DP T cells present as a part of IELs have also their own phenotypes, functionality, and originality. Thus, it will be beneficial to include the full descriptions on diverse features of the DP IELs in the manuscript. Although current main body is with regard to the peripheral mature DP T cells, adding knowledge and understanding of the mature DP IELs would be beneficial for this review article. Second, as described well by authors, the immature DP T cells in the thymus differentiated from double negative (DN) T cells undergo T cell receptor rearrangements for differentiation to single positive T cells. Thus, it is suggestable to revise Figure 1 by adding some more illustrations indicating the DN T cells and the immature DP T cells, so that comprehensive processes of the differentiations to mature heterogeneous DP T cells are displayed. 

I do not feel that English language of overall text in this manuscript requires extensive editing. But, minor checking of the manuscript including typos can be recommended. 

Author Response

We thank Reviewer#3 for her/his constructive suggestions.

We included descriptions of mature DP T cells as part of the intestinal intraepithelial lymphocytes in the 1. Introduction (lines 42-45), 3. Potential origin of mature CD4+CD8+ double-positive T cells (lines 137-144) and 5. Peripheral CD4+CD8+ double-positive T cells sections (lines 230-232).

Further, we improved Figure 1 by adding thymic immature DN and DP T cells. (lines 145-156)

The manuscript was screened for typos and corrected.

Round 2

Reviewer 1 Report

The authors have adequately addressed all the major and minor points, and I have no further comments.

Reviewer 2 Report

Authors significantly improved the manuscript. No more comments.